# A genetic link between discriminative fear coding by the lateral amygdala, dopamine, and fear generalization

**Graham L Jones, Marta E Soden, Cerise R Knakal, Heather Lee, Amanda S Chung, Elliott B Merriam, Larry S Zweifel\***

Department of Psychiatry and Behavioral Sciences and the Department of Pharmacology, University of Washington, Seattle, United States

**Abstract** The lateral amygdala (LA) acquires differential coding of predictive and non-predictive fear stimuli that is critical for proper fear memory assignment. The neurotransmitter dopamine is an important modulator of LA activity and facilitates fear memory formation, but whether dopamine neurons aid in the establishment of discriminative fear coding by the LA is unknown. NMDA-type glutamate receptors in dopamine neurons are critical for the prevention of generalized fear following an aversive experience, suggesting a potential link between a cell autonomous function of NMDAR in dopamine neurons and fear coding by the LA. Here, we utilized mice with a selective genetic inactivation functional NMDARs in dopamine neurons (DAT-NR1 KO mice) combined with behavior, in vivo electrophysiology, and *ex vivo* electrophysiology in LA neurons to demonstrate that plasticity underlying differential fear coding in the LA is regulated by NMDAR signaling in dopamine neurons and alterations in this plasticity is associated non-discriminative cued-fear responses.

**\*For correspondence:** larryz@uw.edu

## Introduction

Across taxa the amygdala is a central locus for fear processing (*Weiskrantz, 1956*; *Goldstein, 1965*; *Slotnick, 1973*; *Adolphs et al., 1995*). Comprised of several interconnected subdivisions that are populated by different types of neurons, the prevailing view is that substructures within the amygdala have specific roles for the acquisition, expression, and extinction of fear-related memories (*LeDoux, 2000*; *Maren and Quirk, 2004*; *Ehrlich et al., 2009*).

Within the amygdala the LA is a major site of convergence for information that arrives from cortical and thalamic nuclei and represents an early processing point for emotionally salient information (*LeDoux, 2000*). Consistent with an early role of the LA in fear memory formation, inactivation of this area prior to, but not after, fear conditioning prevents cued fear behavior (*LeDoux et al., 1990*; *Wilensky et al., 1999*). Optogenetic activation (*Johansen et al., 2010*) or suppression (*Johansen et al., 2014*) of principal neurons in the LA facilitates and impairs cued fear memory formation, respectively. Direct stimulation of principal neuron cell bodies in the basolateral amygdala (BLA) can also increase anxiety-like behavior (*Tye et al., 2011*), thus activation of the LA/BLA is a key neural substrate of fear and anxiety. In addition to principal projection neurons, distinct interneuron populations within the BLA can also potently regulate fear memory formation and fear coding (*Wolff et al., 2014*), indicating that both inhibitory and excitatory synaptic transmission within this region regulates fear-related behavior and learning.

Fear generalization has been proposed to occur through a failure in an animal's ability to define specific outcome contingencies (*Grillon, 2002*; *Lovibond and Shanks, 2002*). Thus, aberrant fear coding in the LA may be an early site of generalized fear manifestation. Consistent with this hypothesis, exposure to fear-inducing stimuli has been demonstrated to increase activity in the

**eLife digest** When we experience a situation that causes us to feel fearful, the brain processes information about the events that led up to it. This information is encoded by groups of nerve cells called neurons in a region of the brain called the lateral amygdala. The nerve cells communicate with each other through chemicals called neurotransmitters. At a junction between two neurons—called a synapse—neurotransmitters are released from one cell and influence the activity of the other cell.

Long-term changes in the strength of these communications in response to specific cues underlie the formation of memories about fearful events. When these changes occur incorrectly they can lead to memories about particular events becoming inaccurate, which can lead to fear being associated with related, but non-threatening, situations. This 'generalization' of fear can lead to generalized anxiety disorder and post-traumatic stress disorder. Dopamine is a neurotransmitter that plays an important role in forming memories of fearful events. However, it is not clear whether neurons that release dopamine are also involved in correctly discriminating fearful events from non-fearful ones.

'Receptor' proteins called NMDARs on the surface of neurons that release dopamine are critical for preventing generalized fear. These receptors detect another neurotransmitter called glutamate. Jones et al. used genetics and 'electrophysiology' techniques to study these receptors in mice. The experiments show that a gene that encodes part of an NMDAR in dopamine neurons plays a key role in how fear memories are formed. When this gene is selectively switched off in the dopamine neurons, mice are more likely to develop generalized fear and anxiety behaviors after a threatening experience.

The experiments also demonstrate that these generalized threat responses are associated with differences in the way the synaptic connections in the lateral amygdala are strengthened. The next major challenge will be to find out which specific synaptic connections are strengthened and to establish how dopamine neuron activity patterns influences this connectivity.

amygdala of human subjects (*Breiter et al., 1996*; *Morris et al., 1996*; *Rauch et al., 2000*), and hyper-amygdala activation is observed in numerous disorders, including post-traumatic stress disorder (*Rauch et al., 2000*). In rodents, increasing the intensity of an unconditioned fear stimulus increases fear generalization that correlates with enhanced activation of LA neurons to both conditioned (CS+) and non-conditioned (CS−) stimuli (*Ghosh and Chattarji, 2015*). In addition, suppression of GABA$_B$-mediated signaling in the LA facilitates non-associative long-term potentiation (LTP) of excitatory synapses that correlates with generalized fear responses (*Shaban et al., 2006*). Collectively, these data suggest that aberrant plasticity in the LA that facilitates non-selective potentiation of excitatory drive or suppression of inhibitory tone is an important neural substrate of generalized fear.

LA neurons demonstrate short latency responses to auditory, visual, and somatosensory stimuli (*Ben-Ari and Le Gal la Salle, 1971*). Following fear conditioning the latency of responses to predictive auditory stimuli decreases (*Quirk et al., 1995*; *Maren, 2000*) and the response amplitude increases selectively to predictive, but not non-predictive, stimuli (*Collins and Pare, 2000*), thus demonstrating acquired selectivity in responding to CS+ and CS− stimuli. Acquisition of differential coding of predictive and non-predictive fear-related information is correlated with studies investigating changes in synaptic strength in the LA after fear conditioning. Auditory fear conditioning elicits LTP-like effects in LA neurons (*Rogan and LeDoux, 1995*; *Rogan et al., 1997*) and elicits presynaptic enhancement of inputs arriving in the LA from the medial geniculate nucleus of the thalamus and cortex (*McKernan and Shinnick-Gallagher, 1997*; *Tsvetkov et al., 2002*; *Zinebi et al., 2002*).

Dopamine is a potent modulator of plasticity in the LA. In the anesthetized rat, odor-evoked potentials in the LA are potentiated following pairing with a footshock that is dependent on dopamine receptor activation (*Rosenkranz and Grace, 2002*). In addition, dopamine signaling modulates local inhibitory networks in the LA (*Loretan et al., 2004*), facilitates LTP induction through suppression of feedforward inhibition (*Bissiere et al., 2003*), and regulates intrinsic excitability of principal neurons (*Yamamoto et al., 2007*). Consistent with the modulation of fear-evoked plasticity in the LA, suppression of dopamine signaling attenuates acquisition of conditioned fear memory (*Borowski and Kokkinidis, 1996*; *Lamont and Kokkinidis, 1998*; *Guarraci et al., 1999*, *2000*; *Greba et al., 2001*) and fear memory expression (*Nader and LeDoux, 1999*).

Subsets of dopamine neurons are activated by aversive stimuli (*Schultz and Romo, 1987*; *Guarraci and Kapp, 1999*; *Brischoux et al., 2009*) and undergo plasticity following an aversive experience (*Lammel et al., 2011*) or fear conditioning (*Guarraci and Kapp, 1999*; *Brischoux et al., 2009*; *Gore et al., 2014*). NMDAR signaling in dopamine neurons regulates both synaptic plasticity (*Bonci and Malenka, 1999*) and phasic activation of these cells (*Overton and Clark, 1997*), and disruption of NMDARs in dopamine neurons of mice results in the development of behavioral correlates of generalized fear and anxiety following an aversive experience (*Zweifel et al., 2011*). These results suggest a potential link between cell autonomous functions of this receptor in dopamine neurons and the disruption of fear coding in the amygdala that may be an underlying cause of fear generalization.

Here we demonstrate that mice lacking NMDARs in dopamine neurons have deficits in cued fear discrimination and that this deficit is associated with altered fear coding in the LA. Contrary to our initial hypothesis, fear generalization in DAT-NR1 KO mice was not associated with a non-specific increased activation of LA neurons to a non-US predictive conditioned stimulus (CS−), but rather a lack of enhanced LA activation to a US predictive conditioned stimulus (CS+). These findings demonstrate that erroneous enhancement of LA activity alone is not necessary for fear generalization, but rather cued fear discrimination coding in general is an essential component for proper fear assignment.

## Results

### Generalized fear in DAT-NR1 KO mice alters signaling in the LA

We have shown previously that DAT-NR1 KO mice develop non-selective enhancement of acoustic startle reflex and increased anxiety-like behavior following an aversive experience (*Zweifel et al., 2011*). To determine whether DAT-NR1 KO mice fail to discriminate between fear predictive and non-predictive stimuli we assayed their performance in a delayed Pavlovian cued fear discrimination task (*Figure 1A*). Mice were presented with 10 trials each of a 10 s light (constant illumination) cue (CS+) that co-terminated with delivery of an unconditioned (US) fear stimulus (0.3 mA, 0.5 s footshock), randomly interspersed with delivery of a distinct light (pulsing illumination) cue (CS−) that did not co-terminate with delivery of the US. 24 hours following conditioning mice were given interspersed presentations of the CS+ and CS− in a novel context. Both DAT-NR1 KO mice and littermate controls displayed enhanced fear (freezing behavior) following both one and two days of conditioning. Control mice demonstrated cue discrimination on both days, a behavior that was significantly impaired in DAT-NR1 KO mice (*Figure 1B*).

The LA is known to participate in cued-fear processing (*LeDoux, 2000*) and altered fear coding in this region correlates with generalized fear responses (*Ghosh and Chattarji, 2015*). Fos expression in the amygdala has been previously shown to be induced following unconditioned footshock and is proposed as an early marker of plasticity (*Campeau et al., 1991*). To determine whether activity-dependent processes in the LA are altered in DAT-NR1 KO mice, we analyzed the induction of the immediately early gene Fos 90 min following one conditioning session in context A (*Figure 1A*). Fos expression was significantly reduced in DAT-NR1 KO mice relative to controls (*Figure 1C,D*) indicating a potential reduced activation of the LA.

### Enhancement of LA activity following US delivery is absent in DAT-NR1 KOs

To determine whether an aversive US results in a general increase in LA activation in a manner that is dependent on NMDAR signaling in dopamine neurons, we recorded the activity of amygdala neurons during three days of fear conditioning following bilateral implantation of tetrodes into the LA of DAT-NR1 KO and control mice (an independent cohort from those used for behavioral analysis). The majority of tetrodes were localized to the ventral lateral and ventral medial subdivisions of the LA (*Figure 2A*). Average waveforms of isolated neurons were similar between groups and across days (*Figure 2B*); similarly, average baseline firing rate and the range of firing rates did not differ between groups or across days of conditioning (*Figure 2C*).

In a number of control neurons we detected an increase in overall firing rate of the cell following presentation of the first US (*Figure 2—figure supplement 1*), with the highest proportion of activated neurons observed in control mice on the first day of conditioning (*Figure 2—figure supplement 1*). To determine whether increased activity occurs more prominently following CS + trials, or represents a general increase, we concatenated all CS + trials with subsequent inter-trial intervals (ITIs;

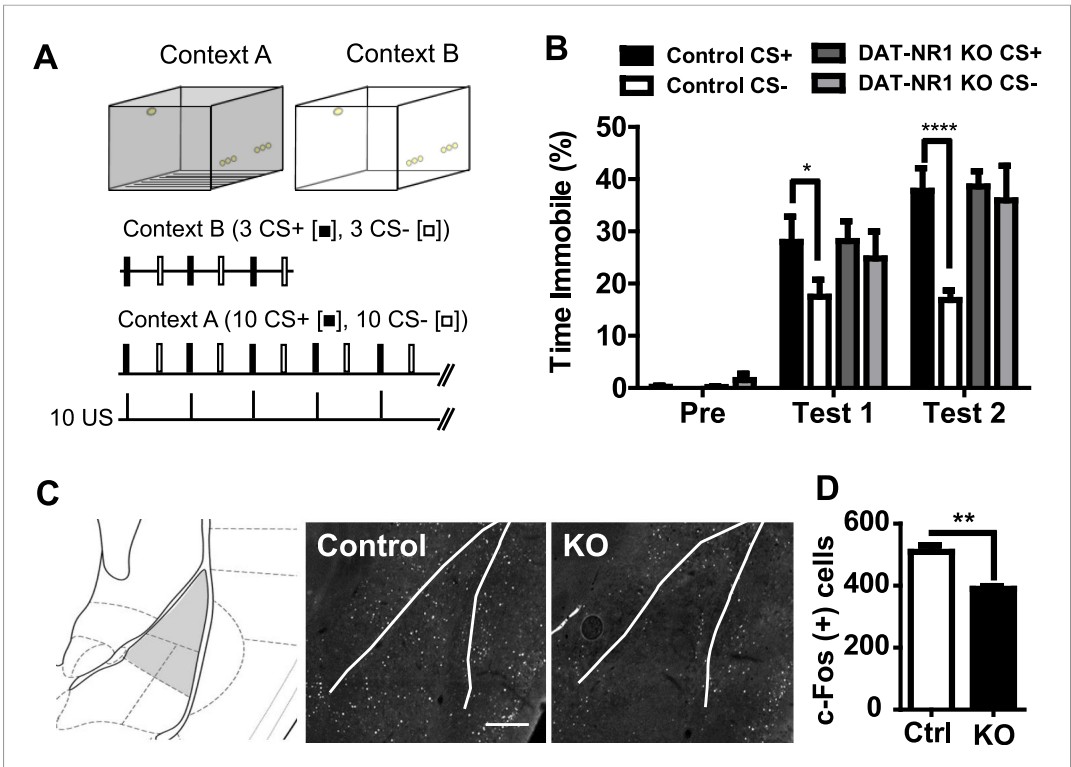

**Figure 1**. Impaired fear discrimination and c-Fos activation in the LA of DAT-NR1 KO mice. (**A**) Fear conditioning paradigm. Mice were probed for freezing in context B (top right) to the CS+ and CS− with three presentations of each stimulus (middle) prior to fear conditioning (Pre) and 24 hr after each conditioning session (Test 1 and Test 2). Mice were conditioned in context A (top left) with 10 presentations of a CS− or CS + co-terminating with US delivery (bottom). (**B**) Freezing behavior (% Time Immobile) during presentation of the CS+ and CS− during pre-conditioning and Test 1 and Test 2. (**C**) Brain atlas image (left) (*Paxinos and Franklin, 2013*) illustrating LA subdivisions (gray shading) analyzed for c-Fos induction following fear conditioning. Representative c-Fos immunoreactive cells in the LA of control (control, left) and DAT-NR1KO (KO, right) mice following a single fear conditioning session. Scale bar = 250 μm (**D**) Average c-Fos positive cells in the LA of Ctrl and KO mice following fear conditioning (n = 3 mice each group, 8 sections/mouse). p < 0.01, unpaired Student's T-test.

*Figure 2D*) and CS− trials with subsequent ITIs (*Figure 2F*) and compared them to baseline activity. In control mice, population activity during both CS+ and CS− trials and their subsequent ITIs was significantly enhanced relative to baseline on the first day of conditioning, but this elevated activity diminished across conditioning days (*Figure 2D–G*). This change in activation was reflected in the proportion of cells activated across days (*Figure 2—figure supplement 1*). We did not detect a significant increase in population activity in DAT-NR1 KO mice (*Figure 2D–G*).

## Enhancement of US activated neurons is absent in DAT-NR1 KO mice

Visual inspection of our population data revealed discrete phasic events time-locked to the presentation of the US; these results are consistent with previous reports of LA neurons responding to footshock (*Ben-Ari and Le Gal la Salle, 1971*; *Romanski et al., 1993*). To determine whether these phasic events undergo plasticity across days of conditioning that is dependent on NMDAR signaling in dopamine neurons, we analyzed neurons with US responsiveness. Action potential waveforms, and baseline firing rates of neurons activated by the US did not differ between genotypes (*Figure 3A,B*) and the proportion of cells responding did not differ between groups or across days (*Figure 3C*). In control mice we found a transient potentiation of the US response, with day 2 responses significantly higher than day 1 (*Figure 3D–F*). This effect was not present on the third day of conditioning. Moreover, we did not detect this effect in DAT-NR1 KO mice (*Figure 3D–F*). The response to the US

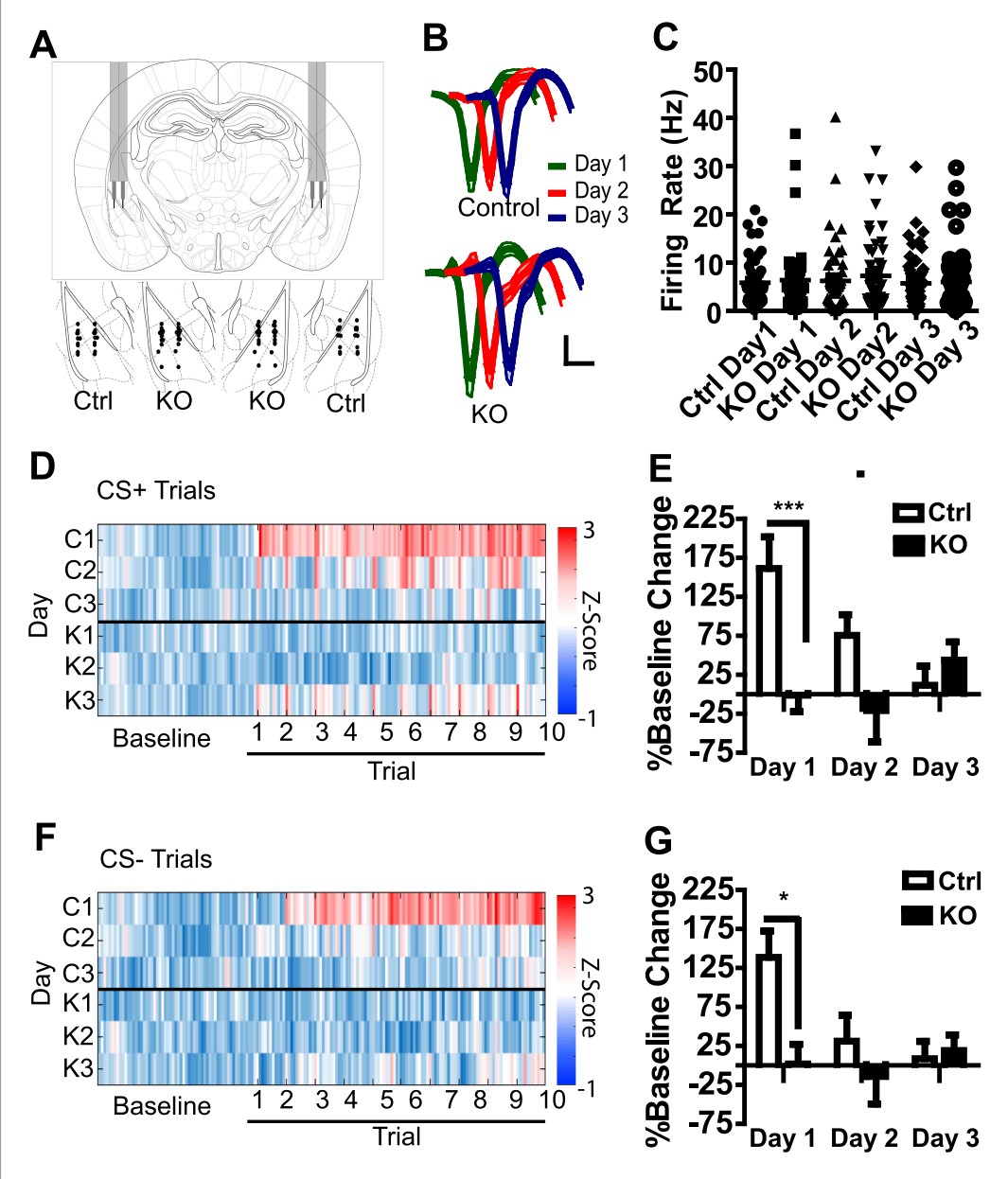

Figure 2. Population activity in the LA is not enhanced in DAT-NR1 KO mice following footshock. (A) Brain atlas image (*Paxinos and Franklin, 2013*) illustrating bilateral tetrode implantation (top) and location of recording electrodes in Ctrl and KO mice. (B) Average waveform of recorded units in Ctrl and KO mice across days of conditioning. (C) Baseline firing rate of individual units in Ctrl and KO mice across days of conditioning (Control: n = 55, 52, and 54 Days 1–3, respectively; DAT-NR1 KO: n = 48, 57, and 58 Days 1–3, respectively). (D) Heat plot of normalized activity in concatenated CS + trials from Ctrl (top) and KO (bottom) mice. (E) Percent change from baseline activity during CS + trials following presentation of the US across days of conditioning. (F) Heat plot of normalized activity in concatenated CS− trials from Ctrl (top) and KO (bottom) mice. (G) Percent change from baseline activity during CS− trials following presentation of the US across days of conditioning. (E, G) Data are presented as the mean ± S.E.M. Repeated measures ANOVA, p < 0.001 and p < 0.05, Bonferroni post-test.

The following figure supplement is available for figure 2:

Figure supplement 1. Population activity during fear conditioning.

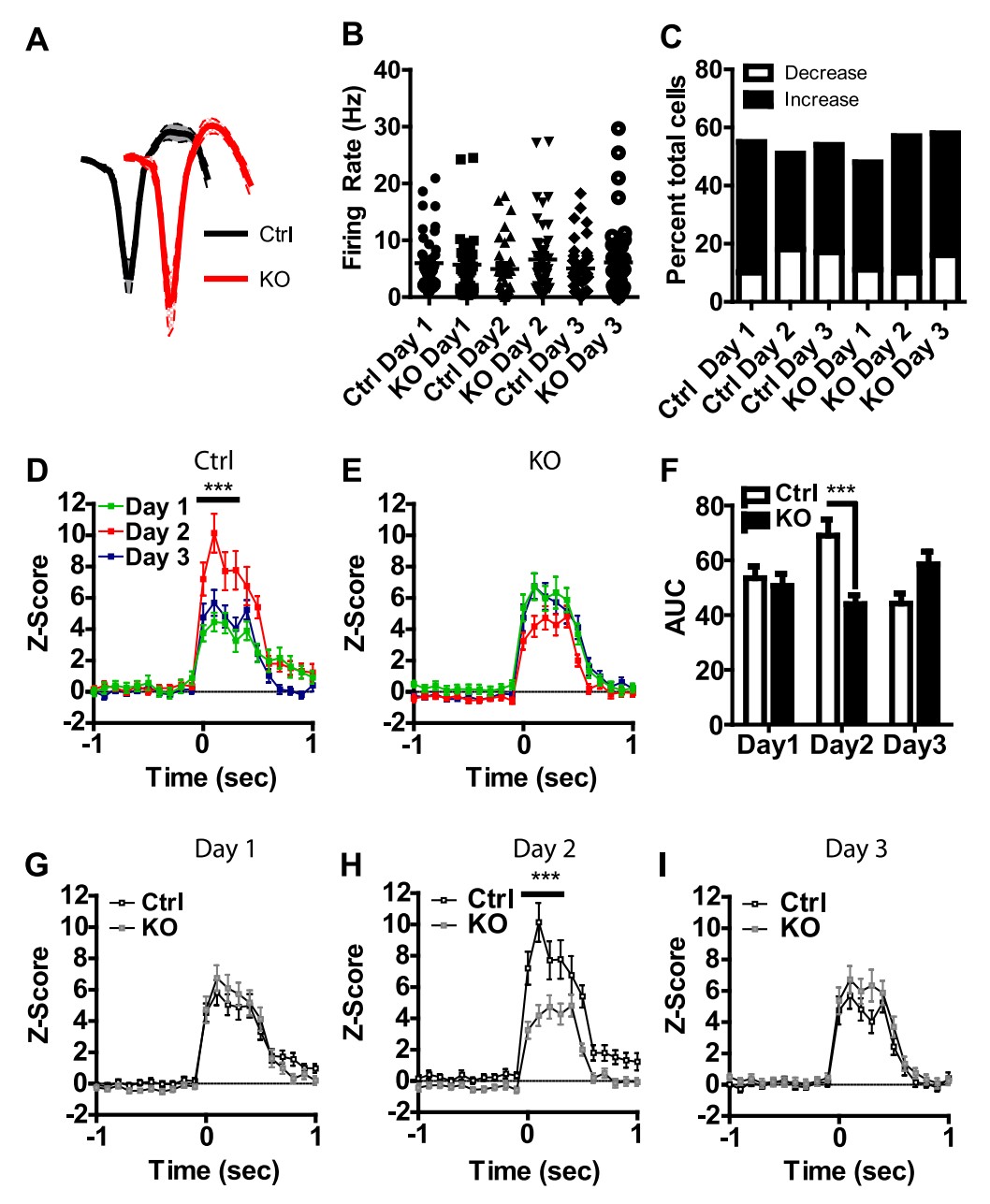

**Figure 3**. Transient plasticity in US-activated LA neurons is absent in DAT-NR1 KO mice. (**A**) Average waveform of recorded units in Ctrl and KO mice that were activated by the US. (**B**) Baseline firing rate of individual units in Ctrl and KO mice that were activated by the US (Control: n = 32, 26, and 28 Days 1–3, respectively; DAT-NR1 KO: n = 25, 34, and 35 Days 1–3, respectively). (**C**) Proportion of neurons from Ctrl and KO mice that were activated or inhibited by the US. (**D**) Average normalized firing rate of US activated neurons in Ctrl mice across days of conditioning. (**E**) Average normalized firing rate of US-activated neurons in KO mice across days of conditioning. (**F**) Average area under the curve (AUC) of activated response for Ctrl and KO mice across days. (**G**–**I**) Comparison of activated responses of Ctrl and KO mice during day 1 (**G**), day 2 (**H**), and day 3 of conditioning (**I**). Data are presented as the mean ± S.E.M. Repeated measures ANOVA, p < 0.001, Bonferroni post-test.

The following figure supplement is available for figure 3:

**Figure supplement 1**. US-inhibited LA neurons do not change across days of conditioning.

on the second day was significantly higher in controls than in DAT-NR1 KO mice (*Figure 3H*), but did not differ on the other days (*Figure 3G,I*).

While the majority of responsive cells increased their firing rates during the US, a small number of neurons showed decreased activity relative to baseline during US presentation (*Figure 3C* and *Figure 3—figure supplement 1*), similar to previous reports (*Ben-Ari and Le Gal la Salle, 1971*). Neurons displaying transient inhibition to the US showed little change across days of conditioning and were not significantly different between genotypes (*Figure 3—figure supplement 1*).

## DAT-NR1 KO mice have impaired discrimination in CS+/CS− activated neurons

Previous studies have shown that LA neurons undergo plasticity in CS−evoked responses, demonstrating increased response magnitude (*Quirk et al., 1995*, *1997*; *Maren, 2000*) and enhanced discrimination between CS+ and CS− stimuli (*Collins and Pare, 2000*; *Ghosh and Chattarji, 2015*). Neurons were detected in both control and DAT-NR1 KO mice that responded to the CS+ and CS− stimuli. Average waveforms, firing rate, and the proportion of neurons responding to these stimuli did not differ between groups (*Figure 4A–C*). In control mice we observed a progressive enhancement of the CS + response in neurons activated by the stimulus across days (*Figure 4—figure supplement 1*). In contrast, DAT-NR1 KO mice did not show enhancement of CS + responses (*Figure 4—figure supplement 1*). Neither control nor DAT-NR1 KO mice showed changes in CS− responses (*Figure 4—figure supplement 1*). Enhanced responding to CS + across days of conditioning in control mice was not due to repeated presentations of the cue, as this effect was not observed in control mice that received three consecutive days of cue (CS+ and CS−) presentation delivered without footshock (unpaired, *Figure 4—figure supplement 1*). Increased CS + responses in control mice across days of conditioning resulted in significant cue discrimination that was absent in DAT-NR1 KO mice (*Figure 4D,E*). This change was not associated with differences in the proportion of neurons responding to both CS+ and CS− stimuli (Control: Day 1, 84%; Day 2, 78%; Day 3, 87% vs DAT-NR1 KO: Day 1, 88%; Day 2, 80%; Day 3, 80%). To determine whether discriminative fear coding emerges early in conditioning, we analyzed the responses of neurons on day 1 of conditioning during the first and last cue presentation. Although there was a trend towards discrimination in both control and DAT-NR1 KO mice by the last conditioning trial this effect did not reach statistical significance (*Figure 4—figure supplement 2*). These findings indicate that early cue discrimination may occur in both groups of mice, but is only maintained in control mice.

Following auditory fear conditioning the latency of responses to a conditioned stimulus decreases in the LA (*Quirk et al., 1995*; *Maren, 2000*). To determine whether the latency of responding to a visually paired stimulus also decreases with fear conditioning we assessed cumulative frequency distributions in the latency to fire across days. We did not observe a significant change in frequency distribution of CS + responses in control mice across days of conditioning (*Figure 4—figure supplement 3*). In contrast, we did observe a significant difference in the distribution in DAT-NR1 KO mice, with increasing response latencies on the third day of conditioning (*Figure 4—figure supplement 3*). Intriguingly, we observed a similar increase in latency across days of conditioning in response to the CS− in both control and DAT-NR1 KO mice (*Figure 4—figure supplement 3*).

## DAT-NR1 KO mice have impaired discrimination in CS+/CS− inhibited neurons

In addition to neurons activated by the CS+ and CS− we also observed a number of neurons that displayed transient inhibitions in response to these stimuli. Average waveforms, firing rate, and the proportion of neurons inhibited by these stimuli were similar across groups (*Figure 5A–C*). Repeated days of conditioning did not alter inhibitory responses to the CS+ in either group (*Figure 5—figure supplement 1*). However, we did observe a progressive decrease in the magnitude of the inhibitory response to the CS− in control mice, with the largest inhibition observed on day 1 and significant reductions in this inhibition on subsequent days (*Figure 5—figure supplement 1*). We did not observe this plasticity in DAT-NR1 KO mice (*Figure 5—figure supplement 1*). Similar to the acquisition of cue discrimination in neurons activated by the CS+ and CS−, control, but not DAT-NR1 KO mice, acquired cue discrimination in neurons inhibited by these stimuli (*Figure 5D,E*). This change was also not associated with differences in the proportion of neurons responding to both CS+ and CS− stimuli

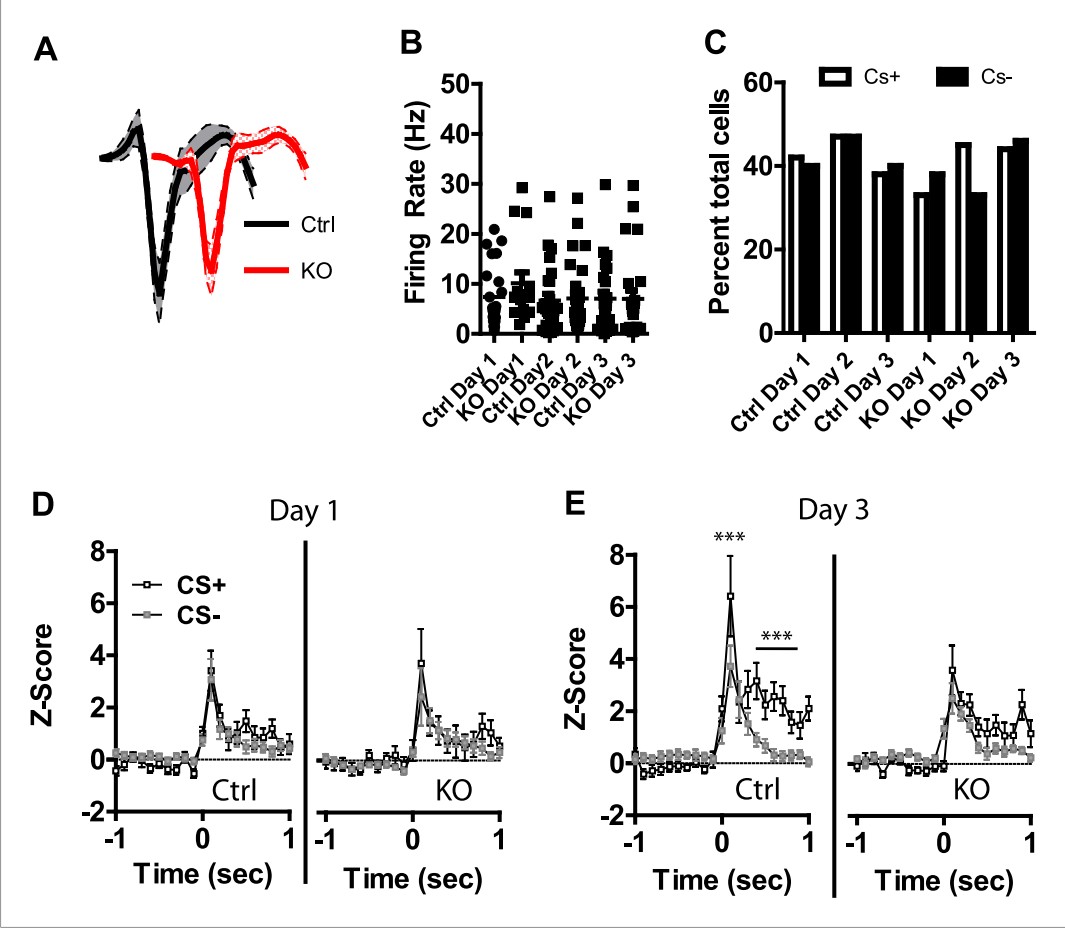

**Figure 4**. Plasticity in CS activated LA neurons is absent in DAT-NR1 KO mice. (**A**) Average waveform of recorded units in Ctrl and KO mice that were activated by the CS+ and CS−. (**B**) Baseline firing rate of individual units in Ctrl and KO mice that were activated by the CS+ and CS− (Control: n = 21, 22, and 19 Days 1–3, respectively; DAT-NR1 KO: n = 14, 23, and 23 Days 1–3, respectively). (**C**) Proportion of neurons from Ctrl and KO mice that were activated by the CS+ and CS−. (**D**) Average normalized firing rate of CS+ and CS− activated neurons in Ctrl and KO mice on day 1 of conditioning. (**E**) Average normalized firing rate of CS+ and CS− activated neurons in Ctrl and KO mice on day 3 of conditioning. Data are presented as the mean ± S.E.M. Repeated measures ANOVA, p < 0.001 and p < 0.01, Bonferroni post-test.

The following figure supplements are available for figure 4:

**Figure supplement 1**. Plasticity in CS activated LA neurons is absent in DAT-NR1 KO mice.

**Figure supplement 2**. CS activated LA neurons are not different at the start of conditioning.

**Figure supplement 3**. Differential response latencies in CS activated neurons between control and DAT-NR1 KO mice.

---

(Control: Day 1, 81%; Day 2, 79%; Day 3, 80% vs DAT-NR1 KO: Day 1, 84%; Day 2, 87%; Day 3, 77%). Similar to neurons activated by the CS+ and CS−, neurons inhibited by these stimuli did not differ during the first or last presentation on the first day of conditioning (*Figure 5—figure supplement 1*).

## Synaptic plasticity following fear conditioning is impaired in DAT-NR1 KO mice

Synaptic plasticity in the LA following fear conditioning has been previously reported, indicating presynaptic enhancement of excitatory synapses from both thalamic and cortical inputs

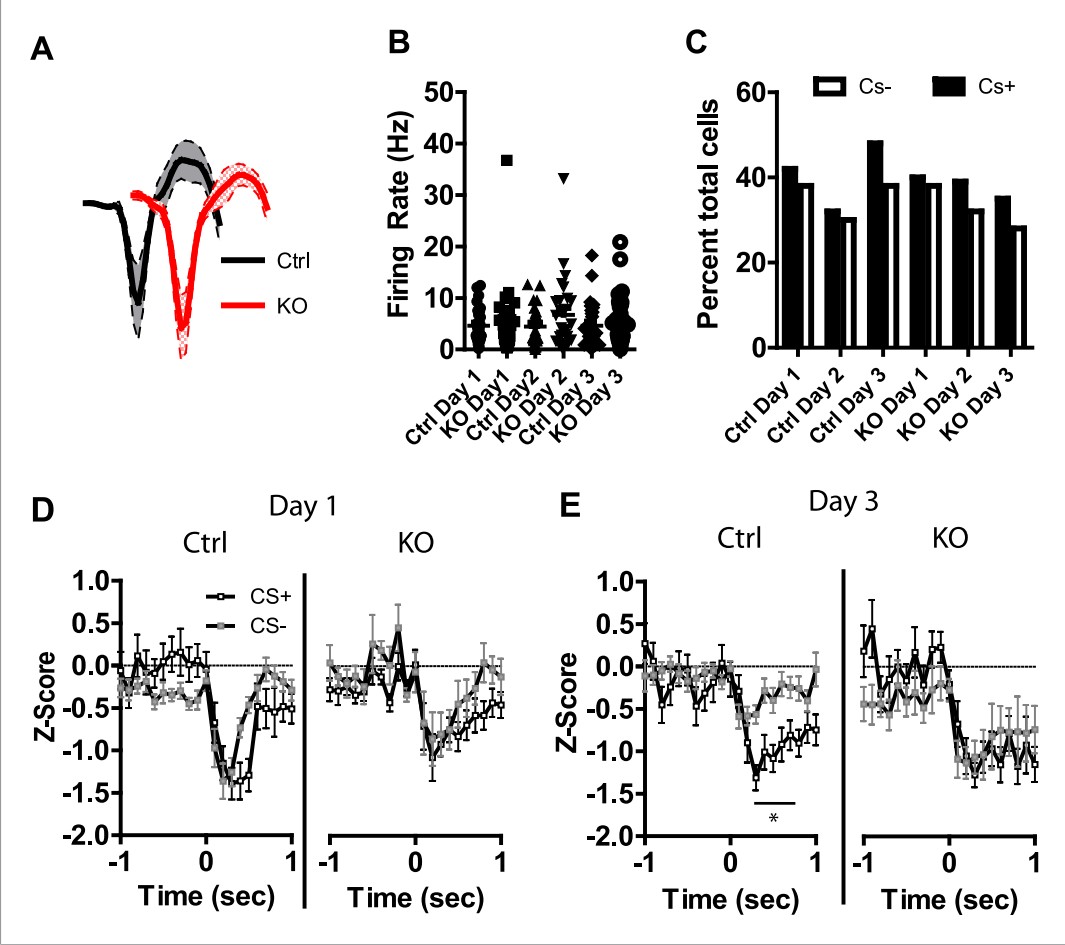

**Figure 5**. Plasticity in CS inhibited LA neurons is absent in DAT-NR1 KO mice. (**A**) Average waveform of recorded units in Ctrl and KO mice that were inhibited by the CS+ and CS−. (**B**) Baseline firing rate of individual units in Ctrl and KO mice that were inhibited by the CS+ and CS− (Control: n = 21, 15, and 22 Days 1–3, respectively; DAT-NR1 KO: n = 13, 20, and 18 Days 1–3, respectively). (**C**) Proportion of neurons from Ctrl and KO mice that were inhibited by the CS+ and CS−. (**D**) Average normalized firing rate of CS+ and CS− inhibited neurons in Ctrl and KO mice on day 1 of conditioning. (**E**) Average normalized firing rate of CS+ and CS− inhibited neurons in Ctrl and KO mice on day 3 of conditioning. Data are presented as the mean ± S.E.M. Repeated measures ANOVA, p < 0.05, Bonferroni post-test.

The following figure supplement is available for figure 5:

**Figure supplement 1**. Plasticity in CS inhibited LA neurons is absent in DAT-NR1 KO mice.

(**McKernan and Shinnick-Gallagher, 1997**; **Tsvetkov et al., 2002**; **Zinebi et al., 2002**). Dopamine receptor signaling has also been demonstrated to influence plasticity in both excitatory and inhibitory pathways in the LA (**Bissiere et al., 2003**; **Loretan et al., 2004**). Our in vivo recordings indicate complex, bidirectional changes in LA cellular activity following multiple days of fear conditioning, including enhanced excitation to the CS+ and reduced inhibition to the CS− in control mice. To determine whether fear generalization associated with disruption of dopamine neuron physiology alters conditioning-evoked plasticity in the LA, we performed whole-cell patch clamp recordings of principal neurons in the LA in acute slices from naïve mice and from mice 24 hr following the second conditioning session.

To test whether fear conditioning elicits pathway-specific changes in the balance of excitatory and inhibitory inputs we measured evoked post-synaptic potentials (PSPs). A stimulating electrode placed in the internal capsule (IC, thalamic), or the external capsule (EC, cortical; **Figure 6A**) reliably evoked a

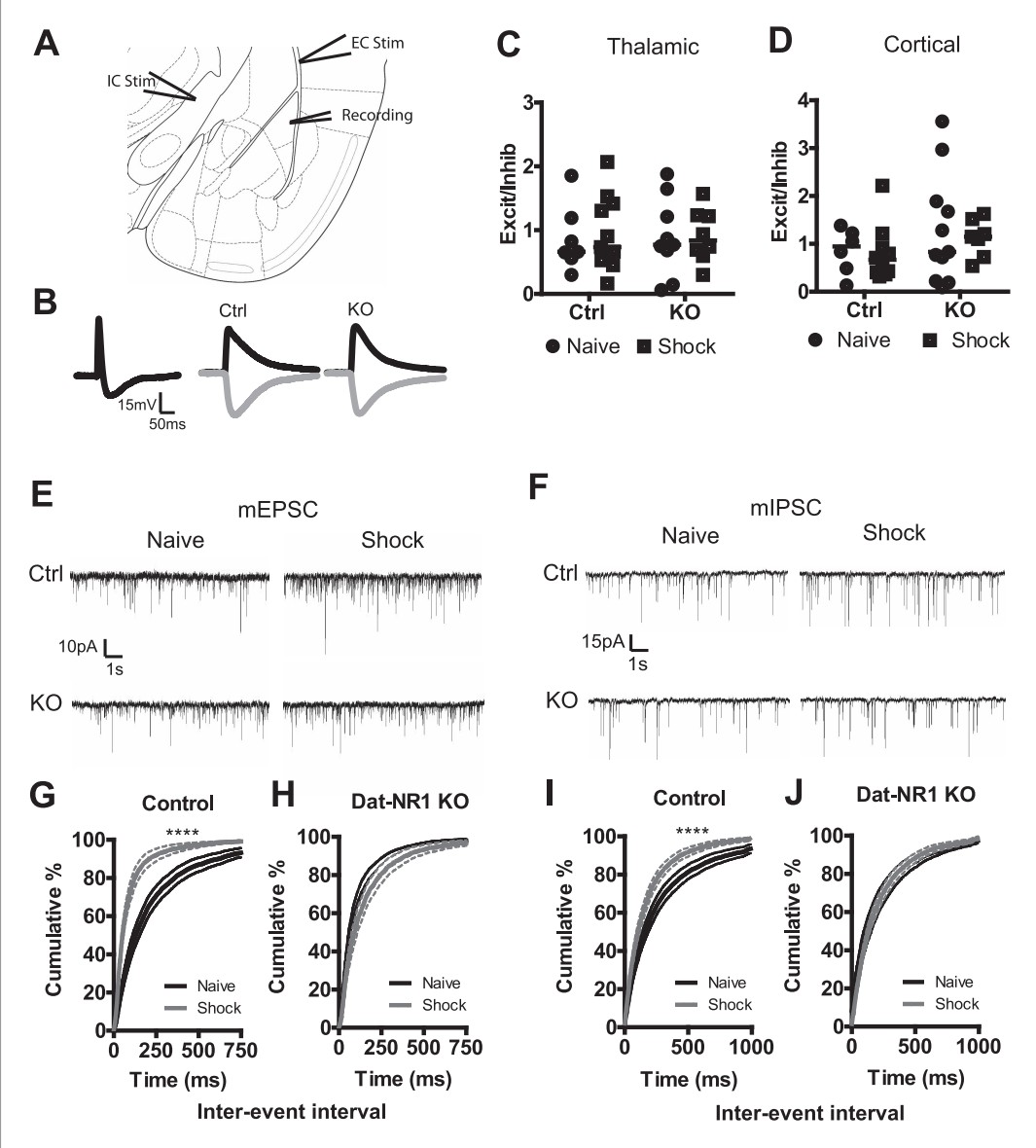

**Figure 6**. Synaptic plasticity in LA neurons is impaired in DAT-NR1 KO mice. (**A**) Brain atlas image[52] illustrating placement of stimulating electrodes in the IC and EC and recording electrode in the LA. (**B**) Representative compound PSP following thalamic stimulation (left) and isolated EPSP and IPSP from Ctrl (middle) and KO (right) mice. (**C**, **D**) Excitation/inhibition ratios of EPSP/IPSPs of individual neurons from Ctrl and KO mice following cortical (**C**, Control: n = 7 naïve, n = 8 shock; DAT-NR1: KO n = 9 naïve, n = 11 shock) and thalamic (**D**, Control: n = 6 naïve, n = 10 shock; DAT-NR1: KO n = 11 naïve, n = 7 shock) stimulations. (**E**, **F**) Representative mEPSCs (**E**) and mIPSCs (**F**) from naïve (black) and fear conditioned (gray) Ctrl and KO mice. (**G**, **H**) Cumulative distribution of mEPSC frequency from naïve and fear conditioned control (**G**, n = 18 naïve, n = 14 shock) and KO mice (**H**, n = 14 naïve, n = 13 shock). (**I**, **J**) Cumulative distribution of mIPSC frequency from naïve and fear conditioned control (**I**) and KO mice (**J**). Data are presented as the mean ± S.E.M. Repeated measures ANOVA, p < 0.0001, Bonferroni post-test.

The following figure supplement is available for figure 6:

**Figure supplement 1**. mEPSC and mIPSC amplitude does not change in LA neurons following fear conditioning.

compound PSP in LA neurons (*Figure 6B*). The excitatory PSP (EPSP) was isolated using bath application of the GABA$_A$ receptor antagonist picrotoxin (100 µM), and the inhibitory PSP (IPSP) was determined through digital subtraction of the EPSP from the compound PSP (*Figure 6B*). We next

calculated the ratio of EPSP:IPSP amplitudes, allowing for the normalized assessment of select changes in either excitatory or inhibitory input. When we compared the ratio of EPSP to IPSP amplitudes in both pathways in both naïve and fear conditioned control and DAT-NR1 KO mice, we saw no significant changes in the balance of excitatory to inhibitory inputs in any group in either the thalamic or cortical pathway (*Figure 6C,D*) indicating a lack of selective change in either inhibitory or excitatory inputs.

To further probe for possible changes in synaptic strength evoked by fear conditioning, we recorded miniature excitatory and inhibitory postsynaptic currents (mEPSCs and mIPSCs) in the LA of both naïve and fear-conditioned mice (*Figure 6E,F*). Fear conditioning evoked a significant increase in the frequency of mEPSCs in control mice, which was not observed in DAT-NR1 KO mice (*Figure 6G, H*). We did not detect significant changes in the amplitude of mEPSCs in either group (*Figure 6—figure supplement 1*). Fear conditioning also elicited a significant enhancement of mIPSC frequency in control, but not DAT-NR1 KO mice (*Figure 6I,J*). We did not detect significant changes in the amplitude of mIPSCs in either group (*Figure 6—figure supplement 1*). These results are consistent with an enhancement of both excitatory and inhibitory inputs to LA neurons.

## Discussion

Our observations that mice lacking NMDARs in dopamine neurons fail to undergo plasticity at multiple levels is consistent with previous reports linking dopamine to plasticity and fear coding in the LA (*Rosenkranz and Grace, 2001*, *2002*; *Bissiere et al., 2003*) and to the regulation of intrinsic excitability of LA neurons (*Yamamoto et al., 2007*). Previous studies have demonstrated that failure of LA neurons to discriminate between fear predictive and non-predictive stimuli is highly correlated with fear generalization (*Ghosh and Chattarji, 2015*). More specifically these authors demonstrate that increased intensity of the US results in non-selective enhancement of LA excitatory responses to a CS−, thus impairing the discrimination between CS+ and CS− (*Ghosh and Chattarji, 2015*). Consistent with these findings it has also been shown that disruption of GABA$_B$ signaling in the LA results in non-associative plasticity, and mice lacking the GABA$_B$ receptor subtype GABA$_{B(1a)}$ develop generalized fear responses (*Shaban et al., 2006*). In contrast to previous reports of generalized fear being associated with an over-exuberance of plasticity and activation of the LA (*Shaban et al., 2006*; *Ghosh and Chattarji, 2015*), in DAT-NR1 KO mice with generalized fear the LA neurons appear as though no CS/US association has been made. More specifically, LA neurons in DAT-NR1 KO mice do not potentiate to the CS+, fail to reduce inhibition to the CS−, and demonstrate longer response latencies to the CS+ and CS−, similar to longer response latencies to the CS− in control mice. Although the manner in which discrimination occurs between activated and inhibited neurons differs, the end result is an acquired differential response that is dependent upon NMDAR signaling in dopamine neurons. These results indicate that generalized freezing behavior to the CS− is not dependent upon excessive activation of the LA, but rather suggest that discriminative coding in the LA reflects the proper assignment of cue-outcome contingencies. Collectively, these findings suggest that disruption of plasticity underlying discriminative fear coding in the LA is associated with generalized fear responses. Our results extend previous findings and confirm that plasticity in the LA is associated with discriminative fear; however, it is not an over-exuberance of plasticity per se that underlies generalization but rather the lack of appropriate plasticity.

If a lack of synaptic strengthening occurs in the LA in the absence of NMDA receptor signaling in dopamine neurons, then where is the site of heightened threat responses following fear conditioning? It is well established that the LA is an early site of convergent sensory information processing that is essential for fear memory acquisition and expression (*LeDoux, 2000*); however, fear coding and plasticity are broadly distributed across multiple brain regions including the thalamus, cortex (sensory and prefrontal), and hippocampus (*Tovote et al., 2015*). In addition, plasticity also occurs within other subdivision of the amygdala including the BLA and lateral subdivision of the central nucleus (CeAl) (*Tovote et al., 2015*). More specifically, generalized defensive responses to threat have been correlated with non-specific coding of cue information in the CeAl (*Ciocchi et al., 2010*), and activation of a specific population of CeAl neurons during discriminative fear conditioning results in response generalization (*Botta et al., 2015*). Thus, it is possible that the generalized threat responses reported here are mediated by exaggerated responses in the CeA or other areas of the brain implicated in fear processing, many of which are innervated by midbrain dopamine neurons.

Consistent with plasticity in the activity of LA neurons in freely moving mice during fear conditioning, we observed synaptic plasticity in LA neurons in acute slice following fear conditioning in control mice that was absent in DAT-NR1 KO mice. Numerous studies have identified pathway-specific plasticity in the LA (*Rogan and LeDoux, 1995*; *McKernan and Shinnick-Gallagher, 1997*; *Rogan et al., 1997*; *Zinebi et al., 2002*; *Schroeder and Shinnick-Gallagher, 2005*; *Shin et al., 2010*; *Cho et al., 2012*). Examination of the ratio of excitatory and inhibitory postsynaptic potentials in response to stimulation of either thalamic or cortical input to LA neurons did not reveal a pathway-specific change in either excitatory or inhibitory drive. These findings indicate that either one or both pathways are enhanced with proportional increases in excitatory and inhibitory drive, or that neither pathway is altered specifically. Our observations of increased frequency of both mIPSCs and mEPSCs suggest a scenario in which one or both pathways are potentiated with proportional changes in excitatory and inhibitory inputs.

There are numerous mechanisms that have been identified that underlie plasticity in LA neurons (*Rodrigues et al., 2004*) with both pre- and postsynaptic sites of action (*Tsvetkov et al., 2002*; *Zinebi et al., 2002*; *Apergis-Schoute et al., 2005*; *Schroeder and Shinnick-Gallagher, 2005*; *Fourcaudot et al., 2008*; *Shin et al., 2010*; *Cho et al., 2012*). Our observed increase in the frequency of both mEPSCs and mIPSCs without a concomitant change in amplitude, is consistent with a presynaptic increase in synaptic transmission, similar to previous descriptions of facilitated synaptic transmission from cortical and thalamic inputs to LA neurons (*McKernan and Shinnick-Gallagher, 1997*; *Tsvetkov et al., 2002*; *Zinebi et al., 2002*). Although we did not observe an increase in mEPSC amplitude this does not exclude postsynaptic changes in glutamate receptors. It was previously demonstrated that trafficking of postsynaptic AMPA receptors in the LA is critical for associative memory formation (*Rumpel et al., 2005*). One potential explanation for these apparent discrepancies is the unmasking of previously silent synapses (*Kerchner and Nicoll, 2008*). Indeed, silent synapses have been reported in the amygdala (*Lee et al., 2013*) and stress has been shown to increase the formation of nascent silent synapses in the LA (*Suvrathan et al., 2014*). Alternatively, it has been proposed that different synaptic sites of LTP in LA neurons (pre- vs postsynaptic) may occur depending on the strength the US, with low US intensities favoring presynaptic LTP and high US intensities favoring postsynaptic LTP (*Shin et al., 2010*). Because we used a low intensity footshock (0.3 mA) it is possible that our protocol favored the induction of a presynaptic form of synaptic strengthening.

The mechanism by which NMDAR signaling in dopamine neurons influences plasticity in fear coding in the LA remains to be elucidated. We have previously demonstrated that NMDARs in dopamine neurons regulate phasic activation of these neurons in response to an aversive stimulus, thus suggesting that phasic dopamine release may facilitate plasticity at excitatory and inhibitory synapses within the LA. Consistent with this hypothesis, previous studies have demonstrated that dopamine signaling within the LA modulates local inhibitory networks (*Loretan et al., 2004*) and gates LTP induction in LA neurons through a suppression of feedforward inhibition (*Bissiere et al., 2003*). In addition to local inhibitory neurons, dopamine neurons innervate paracapsular intercalated cell clusters (*Marcellino et al., 2012*) and potently modulate intercalated neurons through a D1 receptor dependent inhibitory mechanism (*Marowsky et al., 2005*). More specifically, dopamine signaling in lateral paracapsular intercalated neurons suppresses feedforward inhibition from cortical inputs to the BLA (*Marowsky et al., 2005*). Thus, phasic dopamine release facilitated by NMDARs in dopamine neurons is likely to regulate excitability and plasticity of LA neurons through both local inhibitory networks and feedforward inhibition from lateral paracapsular intercalated neurons.

In addition to suppression of inhibition, dopamine increases the excitability of LA neurons through induction of a slow afterhyperpolarization (*Yamamoto et al., 2007*). Collectively, these effects would significantly increase the spike firing of LA neurons. We find that the activity of a significant proportion of LA neurons increases after the first presentation of the US in control mice that parallels our finding of increased Fos levels in the LA following the first conditioning session. Footshock has been demonstrated to increase firing of midbrain dopamine neurons (*Brischoux et al., 2009*), thus increased dopamine release facilitated by NMDAR signaling in dopamine neurons could explain the observed increase in LA activity in control mice that is absent in DAT-NR1 KO mice. Such an increase in the activity of LA neurons would then facilitate the induction of LTP (*Bissiere et al., 2003*) and lasting changes in synaptic strength.

NMDAR signaling in dopamine neurons regulates phasic activation of these neurons as well as synaptic plasticity. Thus, alterations in fear-evoked plasticity within dopamine neurons could also be a major contributor to the disruption of fear coding in the LA of DAT-NR1 KO mice. Using a similar fear

conditioning paradigm we have previously shown that dopamine neurons undergo plasticity in fear-evoked increases in calcium signals in dopamine neurons (*Gore et al., 2014*), which occurs on a similar time course to the results described here. Others have also demonstrated that mesocortical projecting dopamine neurons undergo synaptic plasticity following a painful experience (*Lammel et al., 2011*), indicating pathway specific activation of dopamine neurons. In support of an interdependent plasticity between dopamine neurons and target structures, it has been demonstrated that NMDAR-dependent cocaine-evoked plasticity in dopamine neurons occurs prior to plasticity in dopaminergic targets of the nucleus accumbens and that plasticity within the accumbens is dependent on NMDAR signaling in dopamine neurons (*Mameli et al., 2009*).

The interrelationship between plasticity and phasic activation of dopamine neurons and the relationship between phasic dopamine and plasticity in dopamine neurons that modulates plasticity in the LA remains to be determined. However, our results provide an important first step in linking NMDAR signaling in dopamine neurons with fear discrimination coding in the LA and demonstrate that fear generalization can occur in the absence of hyperexcitation of the LA.

## Materials and methods

### Animals

All Materials and methods were approved by the University of Washington Institutional Animal Care and Use Committee. Control ($Grin1^{\Delta/+}$; $Slc6a3^{Cre/+}$) and DAT-NR1 KO ($Grin1^{\Delta/lox}$; $Slc6a3^{Cre/+}$) mice were generated as previously described (*Zweifel et al., 2008*). 8- to 12-week old male mice were used for behavior and electrophysiology. 5- to 6-week old mice were used for slice electrophysiology. All mice were housed on a 12 hr light/dark cycle in a temperature controlled environment with ad libitum access to food and water for the duration of the study.

### Fear conditioning

Behavioral conditioning was performed in a sound attenuating cabinet with a mouse extra wide modular test chamber outfitted with a shock grid and stimulus lights (Med Associates Inc., St. Albans, VT, United States). Mice were habituated to handling each day one week prior to conditioning. For baseline cue responding mice were placeplaced in the box with a white corrugated plastic insert placed in the chamber to cover the walls and shock grid. Mice were assessed for freezing in response to three randomly interspersed presentations of the CS+ and CS− following a two minute baseline period. Each session of the 2-day fear conditioning paradigm included a 2 min baseline period followed by 20 randomly interleaved trials, comprised of 10 CS + trials and 10 CS− trials, each followed by a 110 s inter-trial interval (ITI). The CS + trials consisted of a constant light cue presentation for 9.5 s, terminating with a 0.5 s 0.3 mA footshock (US) and the CS− trials used a different light cue that flashed 5 times for 200 ms every 2 s, ending with the last light flash and the absence of the US. For assessment of cue-specific freezing behavior 24 hr following conditioning mice were placed in the chamber with the white corrugated plastic insert and freezing was assessed in response to three randomly interspersed presentations of the CS+ and CS− following a 2 min baseline period. Freezing responses were scored as immobility, except for movement associated with breathing, by two independent investigators blind to genotype. Data were analyzed for statistical significance by two-way repeated measures ANOVA.

### In vivo electrophysiology

Electrophysiology in freely moving mice was performed using microdrives fabricated in house, utilizing 16-channel electrode interface boards (EIB-16; Neuralynx) and tetrodes made from 0.00099" diameter tungsten wire (Tungsten 99.95% CS SFV NATRL; California Fine Wire Company). Microdrive implantations in anesthetized mice were stereotaxically targeted at the LA (−1.65 mm A-P, ± 2.85→3.25 mm M-L, −4.3 mm D-V; Paxinos); Bilateral targeting was achieved using coupled polyimide guide tubes (200 µm OD; source) spaced at 6.5 mm. One week after surgery, mice were connected to a 16-channel Medusa Preamplifier and filtered signals (300-5000 Hz) were acquired using a RZ5 Signal Processor (Tucker–Davis Technologies). Tetrodes were lowered daily in ~40 µm increments until unit activity was observed, at which point the animals began the fear conditioning paradigm; tetrodes were not lowered on subsequent days unless cell activity was absent. For conditioning, mice were

treated as above, except they were not exposed to the testing chamber (white corrugated plastic insert) and did not receive cue presentations prior to conditioning. Conditioning proceeded each day following a 10 min baseline period to establish basal neural activity. Tetrode placement was histologically confirmed postmortem using cresyl violet histochemical stain and the presence of implant-induced tissue damage. Neurons were isolated by cluster analysis using Offline Sorter (Plexon) and subsequently formatted and analyzed with MATLAB (Mathworks) and Prism (Graphpad Software). Data were acquired from 13 control and 12 DAT-NR1 KO mice. For unpaired recordings in control mice (N = 5), animals were treated exactly as above, except during the CS + presentation that normally co-terminated with footshock the shock was omitted.

## Unit response determination

Reponses for each cell were normalized to baseline firing rate by calculating a Z-score ($Z = (R_i-R_m)/S.D.$), where $R_i$ = firing rate at an individual time point and $R_m$ = mean firing rate, S.D. = standard deviation of the mean firing rate. Units were characterized as increasing to a stimulus if the Z-score was greater than 1 within the first 500 msec following stimulus presentation. Units were characterized as decreasing to a stimulus if the Z-score was less than −0.5 within the first 500 msec following stimulus presentation. Data were analyzed for statistical significance by two-way repeated measures ANOVA and repeated measures ANOVA, where appropriate.

## Slice electrophysiology

For fear conditioned mice, animals were handled for one week prior to conditioning. Mice were conditioned as above with two consecutive days of 10 CS+/US and CS− presentations. Mice were not pre-exposed to cues prior to conditioning. 24 hours following the second conditioning session mice were euthanized for brain slice preparation. Whole-cell recordings were made using an Axopatch 700B amplifier (Molecular Devices) with filtering at 1 KHz using 4–6 MΩ electrodes. Coronal brain slices (250 μm) were prepared in an ice slush solution containing (in mM): 250 sucrose, 3 KCl, 2 $MgSO_4$, 1.2 $NaH_2PO_4$, 10 D-glucose, 25 $NaHCO_3$, 0.1 $CaCl_2$. Slices recovered for 1 hr at 34°C in artificial cerebral spinal fluid (ACSF) continually bubbled with O2/CO2 and containing (in mM): 126 NaCl, 2.5 KCl, 1.2 $NaH_2PO_4$, 1.2 $MgCl_2$ 11 D-glucose, 18 $NaHCO_3$, 2.4 $CaCl_2$. For evoked PSP and mEPSC recordings patch electrodes were filled with an internal solution containing (in mM): 130 K-Gluconate, 10 KCl, 10 HEPES, 1 EGTA, 5 NaCl, 5 Mg-ATP, 0.5 Na-GTP, 5 QX-314, pH 7.2–7.4, 290 mOsm. For mIPSC recordings patch electrodes were filled with an internal solution containing (in mM): 145 KCl, 10 HEPES, 1 EGTA, 5 Mg-ATP, 0.5 Na-GTP, pH 7.2–7.4, 290 mOsm. ACSF at 32°C was continually perfused over slices at a rate of ∼2 ml/min during recording.

For PSP recordings a concentric bipolar electrode was placed in either the internal (thalamic) or external (cortical) lateral capsule. 1 ms stimuli were delivered at 0.1 Hz, and compound PSPs were recorded in current clamp mode adjusting stimulus intensity until half maximal responses were detected. 15–30 traces were averaged per cell, followed by bath application of picrotoxin (100 μM) to isolate the EPSP. 15–30 EPSP traces were averaged and digitally subtracted from the averaged compound PSP to isolate the IPSP.

mEPSC and mIPSC recordings were made in voltage clamp mode at a holding potential of −60 mV. mEPSCs were recorded in the presence of picrotoxin (100 μM) and tetrodotoxin (500 nM); mIPSCs were recorded in the presence of kynurenic acid (2 mM) and tetrodotoxin (500 nM). Events were detected automatically and were visually inspected and confirmed using Mini Analysis Program (Synaptosoft). Data were analyzed for significance by Student's t-test.

## Histology

For Fos protein analysis, mice were handled and conditioned as above. Mice were euthanized and perfused 90 min following the start of a single conditioning session, which consisted of 10 CS + presentations randomly interspersed with 10 CS− presentations. The CS + co-terminated with a 0.3 mA, 0.5 s footshock (US). Following perfusion, 30 μm frozen sections were collected and incubated overnight at 4° in primary antibody (rabbit anti c-Fos, 1:2000, CalBiochem), then washed and incubated for 1 hr at room temperature in a fluorescently labeled secondary antibody (AlexaFluor488 donkey anti rabbit, 1:250, JacksonImmuno). Sections were imaged using a Nikon Diaphot 200 inverted microscope. c-fos positive cells in the lateral amygdala were counted manually by an

investigator blind to genotype from −1.0 mm posterior to bregma to −2.3 mm posterior to bregma. Cells were counted bilaterally from every fifth section.

## Statistics

Statistical analysis was performed using Prism software (GraphPad).

## Acknowledgements

We thank members of the Zweifel lab for scientific discussion. This work was funded by the US National Institutes of Health (R01-MH094536, LSZ and R21-MH098177, LSZ).

## Additional information

### Funding

| Funder | Grant reference | Author |
| --- | --- | --- |
| National Institutes of Health (NIH) | R01-MH094536 | Larry S Zweifel |
| National Institutes of Health (NIH) | R21-MH098177 | Larry S Zweifel |

The funder had no role in study design, data collection and interpretation, or the decision to submit the work for publication.

### Author contributions

GLJ, MES, Conception and design, Acquisition of data, Analysis and interpretation of data, Drafting or revising the article; CRK, HL, ASC, EBM, Acquisition of data, Analysis and interpretation of data; LSZ, Conception and design, Acquisition of data, Analysis and interpretation of data, Drafting or revising the article, Contributed unpublished essential data or reagents

### Ethics

Animal experimentation: All experimental procedures were performed in accordance with the recommendations in the Guide for the Care and Use of Laboratory Animals of the National Institutes of Health and approved by the University of Washington Institutional Animal Care and Use Committee protocol (#4249-01). All surgical procedures were performed under isolflurane anesthesia with analagesic pretreatment. All efforts were made to minimize suffering.

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
