## [Decision Letter]

Thank you for submitting your work entitled “A Genetic Link Between Discriminative Fear Coding by the Lateral Amygdala, Dopamine, and Fear Generalization” for peer review at *eLife*. Your submission has been favorably evaluated by a Senior editor, a Reviewing editor, and two reviewers.

The reviewers have discussed the reviews with one another and the Reviewing editor has drafted this decision to help you prepare a revised submission.

Summary:

In the current manuscript, Jones and colleagues demonstrate that genetic inactivation of NR1 receptors in dopamine neurons results in aberrant fear discrimination and alterations of neural plasticity in the lateral amygdala, which they suggest may underlie fear generalization. While this is an important step towards elucidating the influence of dopaminergic signaling in discriminative learning, there are a few important points that should be clarified:

Essential revisions:

1) The c-fos experiment is performed after a single conditioning session. To what extent does this correspond to the behavioral procedure used for the subsequent experiments involving in vivo and in vitro recordings, in particular with regard to handling and pre-exposure?

2) The fear discrimination procedure implemented by the authors does not control for the possibility that constant illumination is more readily conditioned than pulsing illumination. Without this control, it remains possible that the impairment of the DAT-NR1 KO mice is simply due to altered sensory processing.

3) In the design of the in vitro recording experiments, the authors draw an equivalence between naïve mice and mice on Day 1 of in vivo recordings; similarly between mice 24 hr following the second conditioning session and mice on Day 3 of in vivo recordings. However, over the course of each conditioning day, mice receive 10 reinforced CS + trials and 10 non-reinforced CS- trials. Therefore, it seems particularly confounded to consider mice on Day 1 of in vivo recordings to be equivalent to naïve mice.

4) In the second paragraph of the Discussion, the authors conclude: “Our observations of increased frequency of both mIPSCs and mEPSCs, along with previous observations of increased excitatory input from both cortex and thalamus following fear conditioning (39, 67, 61), suggest a scenario in which both pathways are potentiated with proportional changes in excitatory and inhibitory inputs.” The authors should therefore provide the data independently for excitatory and inhibitory PSPs in order to substantiate this conclusion.

5) In the third paragraph of the Discussion, the authors write: “In contrast to previous report of generalized fear being associated with an over exuberance of plasticity and activation of the LA (16), in DAT-NR1 KO mice with generalized fear the LA neurons appear as though no CS-US association has been made. More specifically, LA neurons in DAT-NR1 KO mice do not potentiate to the Cs+, fail to reduce inhibition to the CS-, and demonstrate reduced response latencies to the CS+ and CS-, similar to reduced response latencies to the CS- in control mice.” However, this conclusion appears to be based upon the assumption that the Day 1 recordings represent the baseline CS+ and CS- responses, yet over the course of Day 1 the mice receive 10 reinforced CS + trials and 10 non-reinforced CS- trials. Therefore, it would be very helpful to see the baseline (pre-training) CS+ and CS- responses for the reported units in Figures 4 and 5, in order to evaluate whether any plasticity of excitatory and/or inhibitory responses is already evident on Day 1 of conditioning.

6) Do the authors find a correlation between the magnitude of neural responsivity and/or proportion of CS-responsive neurons with the CS-responses of the mice?

7) In Figure 2—figure supplement 1, the authors should provide the proportion of neurons that respond to CS+, CS- and CS+/CS-? Does this differ between control and DAT-NR1 KO mice?

8) Is there a difference in response latencies between control and DAT-NR1 KO mice? In Figure 2—figure supplement 1, there appears to be a shorter response latency for both CS+ and CS- in DAT-NR1 KO mice particularly on Day 3.

---

## [Author Response]

*1) The c-fos experiment is performed after a single conditioning session. To what extent does this correspond to the behavioral procedure used for the subsequent experiments involving* in vivo *and* in vitro *recordings, in particular with regard to handling and pre-exposure?*

The Fos protein level analysis by immunohistochemistry was designed to address the extent to which activity dependent processes are engaged in control and DAT-NR1 KO mice following and initial conditioning session. The mice were treated the same way as animals that underwent fear conditioning for in vivo electrophysiology analysis except that in this assay mice were euthanized 90 min following conditioning. These animals would therefore be most representative of mice recorded during the first day of conditioning. All mice were handled for one week prior to conditioning. One major difference between mice that underwent conditioning for Fos and electrophysiology recordings (both in vitro and in vivo) vs mice that were analyzed for freezing is that mice for Fos and physiology did not receive any pre-exposure to the conditioning chamber. Mice used for Fos and physiology were also not exposed to the testing chamber and were only exposed to the conditioning chamber. Recording during conditioning allowed us to observe neural responses to both the CS and US presentations within the same session. We have included this additional information in our revised Methods section.

2) The fear discrimination procedure implemented by the authors does not control for the possibility that constant illumination is more readily conditioned than pulsing illumination. Without this control, it remains possible that the impairment of the DAT-NR1 KO mice is simply due to altered sensory processing.

We agree that it is possible that the two cues may be differentially conditioned based on whether they were constant or pulsing. It is also possible that this difference contributes to the discrimination deficits in DAT-NR1 KO mice. However, we have previously demonstrated that DAT-NR1 KO mice develop non-specific fear responses in novel contexts and generalized anxiety using a different conditioning paradigm indicating that generalization in these mice is broader than the context used here. In addition, we now report the electrophysiological responses of neurons activated and inhibited by CS+ and CS- stimuli in both control and DAT-NR1 KO mice during the first and last trial of conditioning on the first day. These results reveal that there are no major differences in the response profiles of these neurons on the first day indicating that at least at the level of the LA sensory information is initially processed equivalently. In addition, we also now provide in vivo electrophysiology data from control mice that received the same cues used in conditioning, but were not paired with footshock. This data reveals that response profiles of activated neurons does not change across multiple days of repeated stimulus presentation. Thus our results demonstrating enhanced responding of LA neurons to the CS+ during conditioning reflects associative fear processing and is not related to differential responding to repeated presentation of the different cue stimuli.

*3) In the design of the* in vitro *recording experiments, the authors draw an equivalence between naïve mice and mice on Day 1 of* in vivo *recordings; similarly between mice 24 hr following the second conditioning session and mice on Day 3 of* in vivo *recordings. However, over the course of each conditioning day, mice receive 10 reinforced CS + trials and 10 non-reinforced CS- trials. Therefore, it seems particularly confounded to consider mice on Day 1 of* in vivo *recordings to be equivalent to naïve mice.*

We apologize for the confusion on this matter. We have removed that statement from the manuscript and provided additional detail to the methods to clarify how each group of mice was treated. Although naïve mice were handled the same as conditioned mice they did not receive CS/US pairings. These mice were truly naïve and most closely represented conditioned mice at the start of day one of in vivo electrophysiology recording.

*4) In the second paragraph of the Discussion, the authors conclude: “Our observations of increased frequency of both mIPSCs and mEPSCs, along with previous observations of increased excitatory input from both cortex and thalamus following fear conditioning (*[39]*,*
[67]*,*
[61]*), suggest a scenario in which both pathways are potentiated with proportional changes in excitatory and inhibitory inputs.” The authors should therefore provide the data independently for excitatory and inhibitory PSPs in order to substantiate this conclusion.*

We agree that we cannot support this conclusion based on the mEPSC and mIPSC frequency. We have removed this statement and provided a much more in-depth discussion of these findings. With regard to the quantification of the PSPs, we cannot provide an accurate individual average of the IPSP or EPSPs due to variability in stimulating electrode placement. Typically PSPs are normalized to baseline responses to allow for comparisons across slices and across animals. Because we are comparing two independent groups in which baseline responses cannot be established, an alternative method of normalization is required. To achieve this we compared the ratio of EPSPs to IPSPs, similar to what is done for the generation of AMPA/NMDA receptor ratios. Because we did not see selective effects on EPSPs or IPSPs in either pathway, we analyzed mEPSCs and mIPSCs which do allow for more accurate direct comparisons across groups. In the latter instance we observed and increase in the frequency of both mEPSCs and mIPSCs indicating that both excitatory and inhibitory synaptic inputs are changed in control mice, but not DAT-NR1 KO mice.

*5) In the third paragraph of the Discussion, the authors write: “In contrast to previous report of generalized fear being associated with an over exuberance of plasticity and activation of the LA (*[16]*), in DAT-NR1 KO mice with generalized fear the LA neurons appear as though no CS-US association has been made. More specifically, LA neurons in DAT-NR1 KO mice do not potentiate to the Cs+, fail to reduce inhibition to the CS-, and demonstrate reduced response latencies to the CS+ and CS-, similar to reduced response latencies to the CS- in control mice.” However, this conclusion appears to be based upon the assumption that the Day 1 recordings represent the baseline CS+ and CS- responses, yet over the course of Day 1 the mice receive 10 reinforced CS + trials and 10 non-reinforced CS- trials. Therefore, it would be very helpful to see the baseline (pre-training) CS+ and CS- responses for the reported units in*
Figures 4 and 5*, in order to evaluate whether any plasticity of excitatory and/or inhibitory responses is already evident on Day 1 of conditioning.*

We agree that mice receiving CS/US presentations on day 1 are not equivalent to naïve mice and we have modified previous statements. We have also clarified the text to better convey that mice that underwent fear conditioning to assess freezing are not the same mice that were used for in vivo recordings, or slice recordings. Unfortunately, we did not record baseline cue responses prior to conditioning in mice that underwent in vivo recordings. However, we did investigate the initial responses to both the CS+ and CS- cue presentations in both groups (Day 1, trial 1) and compared those responses to the last trial of the day (Day 1, trial 10). We did not observe significant differences in the responses of either group to the CS+ and CS- cue presentation on the first vs last trial. This data is now included in Figure 4—figure supplement 2.

6) Do the authors find a correlation between the magnitude of neural responsivity and/or proportion of CS-responsive neurons with the CS-responses of the mice?

As mentioned above, we did not perform behavioral analysis on mice undergoing in vivo recordings. We find that freezing behavior in mice implanted with microdrives and tethered to pre-amplifiers is unreliable. For this reason we assessed behavior in independent cohorts of mice and cannot directly correlate unit responses to freezing behavior.

*7) In*
Figure 2—figure supplement 1*, the authors should provide the proportion of neurons that respond to CS+, CS- and CS+/CS-? Does this differ between control and DAT-NR1 KO mice?*

We have now included the proportion of neurons that respond to both CS+ and CS- stimuli across days of conditioning within the manuscript text. We did not observe differences in this proportion across days or across groups, with the majority showing responses to both cues (∼80%).

*8) Is there a difference in response latencies between control and DAT-NR1 KO mice? In*
Figure 2—figure supplement 1*, there appears to be a shorter response latency for both CS+ and CS- in DAT-NR1 KO mice particularly on Day 3.*

We have included the analysis comparing the response latencies to the different stimuli on days 1 and 3 of conditioning. We observed a significant difference in response latency in control vs DAT-NR1 KO mice on Day 3 of conditioning, with controls displaying significantly shorter latencies that DAT-NR1 KO mice.